# Impacts of Mutations in the P-Loop on Conformational Alterations of KRAS Investigated with Gaussian Accelerated Molecular Dynamics Simulations

**DOI:** 10.3390/molecules28072886

**Published:** 2023-03-23

**Authors:** Shuhua Shi, Linqi Zheng, Yonglian Ren, Ziyu Wang

**Affiliations:** 1School of Science, Shandong Jianzhu University, Jinan 250101, China; 2Shandong Key Laboratory of Biophysics, Institute of Biophysics, Dezhou University, Dezhou 253023, China

**Keywords:** KRAS, gaussian accelerated molecular dynamics, G12 mutations, principal component analysis, free energy landscape

## Abstract

G12 mutations heavily affect conformational transformation and activity of KRAS. In this study, Gaussian accelerated molecular dynamics (GaMD) simulations were performed on the GDP-bound wild-type (WT), G12A, G12D, and G12R KRAS to probe mutation-mediated impacts on conformational alterations of KRAS. The results indicate that three G12 mutations obviously affect the structural flexibility and internal dynamics of the switch domains. The analyses of the free energy landscapes (FELs) suggest that three G12 mutations induce more conformational states of KRAS and lead to more disordered switch domains. The principal component analysis shows that three G12 mutations change concerted motions and dynamics behavior of the switch domains. The switch domains mostly overlap with the binding region of KRAS to its effectors. Thus, the high disorder states and concerted motion changes of the switch domains induced by G12 mutations affect the activity of KRAS. The analysis of interaction network of GDP with KRAS signifies that the instability in the interactions of GDP and magnesium ion with the switch domain SW1 drives the high disordered state of the switch domains. This work is expected to provide theoretical aids for understanding the function of KRAS.

## 1. Introduction

The oncogenic family of RAS genes has been paid increasing interest in regards to the treatment of human cancers upon the frequency of activating mutations [1,2]. Rat sarcoma (RAS) proteins, containing harvey-RAS (HRAS), kirsten RAS (KRAS), and neuroblastoma-RAS (NRAS) function as molecular switches that are responsible for regulations on cell growth, apoptosis, and metabolism. As a guanine nucleotide-binding protein, KRAS plays a vital role in growth-promoting signal-transduction pathways by virtue of interconversion between the GTP-bound active state and the GDP-bound inactive one [3,4]. Conversion from GDP to GTP-bound state is achieved by binding of growth factors to extracellular receptors, which induces nucleotide exchange with the help of guanine nucleotide exchange factors (GEFs), such as the son of sevenless (SOS) [5,6]. To the contrary, GTP is slowly hydrolyzed into GDP by KRAS, and this hydrolysis reaction can be greatly accelerated by GTPase activating proteins (GAPs) [7,8,9]. The studies indicated that the presence of mutations in almost all major cancers makes RAS proteins a significant therapeutic target, in particular for KRAS, because it was recognized as one of the most frequently mutated oncogenes [10,11,12,13,14,15]. Thus, it is of importance to further study mutation-induced changes in conformations and activity of KRAS for drug design toward the RAS family.

The previous reports suggested that mutations occurring at codons 12, 13, and 61 of RAS proteins account for 98% of mutations detected in human cancers [10,16,17,18,19]. As a common phenomenon, mutations highly impact conformational alterations of the switch domains in KRAS, composed of the switch domain 1 (SW1, including residues 30–40) and the switch domain 2 (SW2, including residues 59–75) [20,21,22,23,24,25], which is shown in Figure 1A. The crystal structures solved by Buhrman and the coworkers indicated that Q61 mutants greatly affected the conformational states of the switch domains and changed interaction networks of ligands with RAS proteins [26]. Chen et al. performed GaMD simulations on the GTP-bound WT and Q61 mutated KRAS, and their results unveiled that Q61 mutants led to the conformational rearrangement of the switch domains and affected the activity of KRAS [27]. Several key observations suggested that G12 mutants not only altered conformational dynamics of the switch domains, but they also had unique biologic and clinical behaviors [8,20,28,29,30,31]. The experimental structures and computational studies stemming from multiple work teams suggested that G13D mutants in the three isoforms (HRAS, KRAS and NRAS) had an altered active site and conformational changes of the switch domains that affect the stability of the nucleotide-binding pocket [8,32,33,34]. More previous studies revealed that mutations different from codons 12, 13, and 61 also exerted significant effect on the conformation states of the switch domains and the nucleotide binding [12,35,36,37,38,39,40,41,42]. Despite these successes, molecular mechanism of mutation-induced conformational state changes of the switch domains from KRAS are still insufficient currently. Therefore, it is highly requisite to further explore conformational changes of KRAS for advancing the understanding of the structural basis for the function of KRAS.

Compared to the static information arising from the experiments, internal dynamics of conformational alterations of targets plays a more important role in detection of binding sites or pockets. Conventional molecular dynamics (cMD) simulations have been extensively applied to probe conformational dynamics of targets upon ligand bindings and point mutations [43,44,45,46,47,48,49,50,51]. Binding free energy predictions based on cMD trajectories were also utilized to clarify energetic basis of ligand–target identification [52,53,54,55]. More interestingly, cMD simulations have been adopted to successfully probe conformational changes of RAS proteins and understand the effect of mutations on activity of RAS proteins [28,56,57,58,59,60,61,62]. To better improve conformational sampling of targets, accelerated molecular dynamics (aMD) [63] and Gaussian accelerated molecular dynamics (GaMD) [64,65,66,67] simulations were proposed to avoid the possibility of local minimization for conformational sampling and to obtain full conformational relaxation. Recently, great successes of aMD and GaMD simulations have been obtained as insights into the conformational transition and activity regulation of RAS proteins [68,69] and the other targets [70,71,72,73,74,75,76]. The aforementioned description indicated that aMD and GaMD simulations are reliable approaches for deep investigation of molecular mechanism underlying the mutation-mediated conformational transformation of RAS.

To advance the understanding on effect of mutations in codon 12 from the P-loop on conformational states of the switch domains, the GDP-bound wild-type (WT) and G12A, G12D, and G12R KRAS were chosen for the current study. The mutation G12A brings an alkyl group in the side chain and increases the length of the side chain, which changes the hydrophobicity of residue 12. The mutation G12D brings a carbonyl group in the side chain and adds a net negative charge, which alters the polarity around residue 12. The mutation G12R enlarges the size of the side chain and brings a net positive charge in its side chain, which generates effect on the polarity near residues 12. The structural details on G12 mutations and GDP were, respectively, depicted in Figure 1C,D. The changes in the hydrophobicity or polarity caused by G12 mutations certainly affect interactions with GDP and alter the conformation of KRAS. More importantly, these three mutations are involved in the development of human oncology. Thus, it is of great significance to probe molecular mechanism underlying mutation-mediated effect on the conformational transition of KRAS for drug design. To achieve this aim, GaMD simulations, principal component analysis (PCA) [77,78,79,80], construction of free energy landscapes (FELs), and dynamic cross-correlation map (DCCM) calculations [81,82,83] were coupled together to perform this current study.

## 2. Results and Discussion

### 2.1. Structural Fluctuations and Internal Dynamics of KRAS

In order to understand the effect of G12 mutations on structural stability, root-mean-square deviations (RMSDs) of non-hydrogen atoms from KRAS relative to the initially minimized structures were calculated. The time evolution of RMSDs for four systems in three independent GaMD simulations was depicted as Appendix A. The fluctuation ranges of RMSDs for the GDP-bound WT, G12A, G12D, and G12R KRAS are 1.38–3.68, 1.76–4.01, 1.37–4.52, and 1.57–3.45 Å, respectively, suggesting that G12 mutations affect the structural stability of KRAS. As observed at the probability distribution of RMSDs (Figure 2A), the RMSDs of the GDP-bound G12A, G12D, and G12R KRAS are increased by 0.16, 0.4, and 0.4 Å, separately, implying that three mutations enhance structural fluctuations of KRAS compared to the WT KRAS. To clarify the stability of GDP in the binding pocket of KRAS, the RMSDs of non-hydrogen atoms from GDP were computed by referencing the initially minimized conformation, and its time evolution and probability distribution were individually displayed in Appendix A and Figure 2B. The RMSDs of GDP in four systems fluctuate in a range from 0.18 to 2.88 Å, suggesting that GDP is stably kept at the binding pockets of the WT and mutated KRAS (Appendix A). By comparison with the WT KRAS, the RMSDs of GDP in the G12A and G12D KRAS are raised by 0.14 and 0.34 Å, respectively, showing that G12A and G12D strengthen the structural fluctuations of GDP relative to the WT system. Thus, these two mutations correspondingly decrease the structural stability of GDP in the binding pocket of KRAS, in particular for G12D (Figure 2B). The RMSDs of GDP in the WT and G12R KRAS are distributed at the same position of 0.66 Å. However, the probability of RMSD for GDP in the G12R KRAS is much lower than that in the WT one. Moreover, its distribution shape in the G12R KRAS is wider than that in the WT one. Thus, G12R abates the stability of GDP in the binding pocket of KRAS.

To examine if G12 mutations impact the structural flexibility of KRAS, root-mean-square fluctuations (RMSFs) were estimated by using the coordinates of the Cα atoms recorded at the SGT (Appendix A). It was found that the switch domains SW1 and SW2 and the loop L1 are greatly flexible in four systems. The difference in the RMSFs between the WT and mutated KRAS was calculated by using the equation $∆RMSF={RMSF}_{mutation}-{RMSF}_{WT}$, in which ∆RMSF, ${RMSF}_{mutation}$, and ${RMSF}_{WT}$ represent the RMSF difference, as well as the RMSFs of the mutated and WT KRAS (Figure 2C). The results verify that G12A, G12D, and G12R strengthen the structural flexibility of the switch domains SW1 and SW2 relative to the WT KRAS. Moreover, the influence of three mutations on the SW2 is stronger than SW1 (Figure 2C). To the contrary, G12A, G12D, and G12R weaken the structural flexibility of the loop L3 by referencing the WT KRAS (Figure 2C).

To check the effect of mutations on the solvent-accessible extent of KRAS, molecular surface areas (MSAs) of the GDP-bound WT G12A, G12D, and G12R KRAS were estimated based on the SGT by using the linear combination of pairwise overlap (LCPO) method [84] (Figure 2D). Compared to the GDP-bound WT KRAS, D33K hardly changes the MSA of KRAS. The peaks of MSAs for the GDP-bound G12A and G12D KRAS are increased by 124 Å2, implying that G12A and G12D expand the contacting extent of KRAS with the solvent relative to WT KRAS. However, the MSA of the GDP-bound G12R KRAS is reduced by 124 Å2, indicating that G12R leads to the contracting in the KRAS-solvent contacts. To understand the impact of G12 mutations on the compact extent of KRAS, radius of gyrations (Rgs) were computed for the GDP-bound WT, G12A, G12D, and G12R KRAS (Appendix A). The compact extent of the GDP-bound G12A and G12D KRAS is slightly decreased compared to the WT KRAS, while that of the GDP-bound G12R KRAS is slightly increased.

With an expectation to probe mutation-mediated influence on internal dynamics of KRAS, DCCMs were estimated by utilizing the coordinates of the Cα atoms saved at the SGT (Appendix A). The results display that three G12 mutations exert different effects on the internal dynamics of KRAS. For the WT KRAS, the regions R1, R2, and R3 produce slightly anti-correlated motions, in which the region R1 describes the anti-correlated motions between the switch domain SW1 and the P-loop, and the region R2 reflects the anti-correlated movement of the switch domain SW2 relative to the P-loop, and the region R3 characterizes the anti-correlated motion between the loop L3 and the switch domain SW2 (Figure 1A and Appendix A). The region R4 in the GDP-bound WT KRAS generates the positively correlated movement of the loop L2 relative to the P-loop (Figure 1A and Appendix A). By referencing to the WT KRAS, G12A and G12D slightly enhance the anti-correlated motion between the SW1 and the P-loop, while G12R obviously strengthens this anti-correlated motion (Appendix A). By comparison with the WT KRAS, G12D slightly weakens the anti-correlated movement of the SW2 relative to the P-loop (Appendix A), but G12A and G12R evidently raise this anti-correlated movement (Appendix A). Compared to the WT KRAS, G12A and G12R strengthen the anti-correlated motion between the loop L3 and the SW2, while G12D hardly impacts this anti-correlated motion (Appendix A). It was also observed that three G12 mutations slightly enhance the positively correlated movement of the loop L2 relative to the P-loop compared to the WT KRAS.

Based on the aforementioned information, three G12 mutations change structural fluctuations and affect structural stability of KRAS. Meanwhile, they also obviously impact structural flexibility and internal dynamics simulations of the switch domains, as well as solvent-accessible extents of KRAS. In fact, the switch domains not only interact with the nucleotides, but they also mostly overlap with the binding regions of KRAS related to its effectors. Therefore, the changes in the structural flexibility and internal dynamics of the switch domains induced by three G12 mutations certainly influence KRAS-effector binding. It is well known that high flexibility is a main feature of the switch regions of RAS proteins, which enables conformational transformation associated with a GDP/GTP exchange [31]. Thus, the changes in the flexibility of the switch domain SW 1 can be applied to regulate the activity of KRAS. Similar phenomena and findings were also observed at the previous studies [56,69,85,86], supporting this current work. Binding of small molecule inhibitors to an allosteric position of KRAS can lead to conformational alterations of the switch domains from KRAS and change structural flexibility and the activity of KRAS, which provides a hint for future drug design toward RAS proteins.

### 2.2. Free Energy Profile of Mutation-Induced Conformational Transitions of KRAS

FELs are usually adopted to explore the thermodynamics and kinetics of the ligand–receptor and receptor–solvent systems at certain conditions [87]. To decipher free energy profiles correlating with conformational transitions of KRAS, FELs were constructed by using RMSDs of the non-hydrogen atoms from KRAS and the distance of the Cα atom in Y32 away from that in Q61 as reaction coordinates (RCs). The previous analyses revealed that G12A, G12D, and G12R produce different effects on the total structural fluctuations of KRAS, thus RMSDs can reflect the total structural stability. Y32 is situated at the switch domain SW1, while Q61 is located at the SW2. Therefore, the distance between the two residues can rationally capture the conformational transitions of the switch domains. These two facts are the main reason why we selected the RMSDs and the distance as RCs. The FELs and the corresponding structural information were exhibited in Figure 3, Figure 4, Figure 5 and Figure 6.

In the GDP-bound WT KRAS, GaMD simulations capture two free energy basins EB1and EB2 (Figure 3A), implying that the GDP-bound WT KRAS comes across two main subspaces. The distance between the Cα atoms of Y32 in the SW1 and Q61 in the SW2 are 16.7 and 14.6 Å in the structure EB1 and EB2 (Figure 3B,C), respectively, indicating that the conformational space of the switch domains in the WT KRAS does not generate big changes. As displayed in structural superimposition of KRAS in EB1 and EB2, the SW1 hardly deviates from each other in the EB1 and EB2 state, but the loop L1 from the SW2 produces obvious deviation (Figure 3D). In addition, the loop L4 and the helix α4 also yield evident deviations in the GDP-bound WT state (Figure 3D). According to the structural alignment of GDP and magnesium ions (Mg2+) in the structures EB1 and EB2 (Appendix A), GDP and Mg2+ agree well with each other, verifying that these two ligands are kept at the binging pocket through the entire GaMD simulations. To check the difference in the GDP-KRAS interaction in two states EB1 and EB2, the protein–ligand interaction profiler (PLIP) server [88,89] was utilized to analyze the interaction network of GDP with KRAS (Appendix A). The salt bridge interaction of GDP with K16 and electrostatic interaction (EI) of GDP with D119 appear at two states, EB1 and EB2, and hydrogen bonding interactions (HBIs) of GDP with G13, V14, G15, K16, S17, A18, D30, N116, K117, S145, A146, and K147 are detected at two structures, EB1 and EB2 (Appendix A). The only difference is that the HBI of GDP with A11 appearing at the structure EB1 disappears at the structure EB2. In addition, a sodium ion appears near the phosphate group of GDP in two structures, EB1 and EB2 (Figure 3B,C), which possibly provides a compensation for the polarity changes caused by conformational alterations of KRAS.

In the GDP-bound G12A KRAS, GaMD simulations detect four free energy basins EB1-EB4 (Figure 4A), signifying that the GDP-bound G12A KRAS is populated at four main conformational subspaces. The distances of the Cα atom in Y32 away from that in Q61 are 24.1, 20.5, 18.3, and 14.2 Å in the structures EB1, EB2, EB3, and EB4 (Figure 4B–E), respectively, in which the switch domains are most loosely packed in the structure EB1, while they are most tightly packed in the structure EB4. By comparison with the WT KRAS, G12A produces influences on the compact extent of the switch domains. As exhibited at the structural alignment of the GDP-bound G12A KRAS (Figure 4F), the switch domains display a highly disordered state. Furthermore, the disordered extent of the SW2 is higher than that of the SW1. Compared to the WT KRAS, G12A induces more energetic states and results in more unordered situations of the switch domains (Figure 4F). In spite of this, the structures of GDP and magnesium ions agree well with each other in four states, EB1-EB4 (Appendix A), which suggests the high stability of GDP and magnesium ion through the GaMD simulations. The structures EB1 and EB4, respectively, located at the most incompact and tightest states of the switch domains, were adopted to analyze interactions of GDP with KRAS using the PLIP sever (Appendix A). Both the salt bridge interaction of GDP with K16 and EI of GDP with D119, as well as the HBIs of GDP with residues G13, V14, G15, K16, S17, A18, N116, K117, A146, and K147, were detected in two structures at the EB1 and EB4. Differently, the π–π interaction of GDP with F28 and the HBIs of GDP with A11 and S145 appearing at the EB1 disappear at the EB4, while the HBI interactions of GDP with V29 and D30 observed at the EB4 loses at the EB1 (Appendix A), which implies the effect of G12A on the GDP-KRAS interaction network. Besides, a sodium ion (Na^+^) is identified near the phosphate group of GDP in the structure EB3 (Figure 4D). In the structure EB3, the SW 1 goes close to the phosphate group of GDP, which strengthens the electrostatic repulsive interaction between this group in GDP and residues with negative charges in the SW 1. Therefore, the presence of Na^+^ relieves the change in the polarity near the phosphate group of GDP.

For the GDP-bound G12D KRAS, four energy basins (EB1-EB4) are acquired during GaMD simulations (Figure 5A), suggesting that the GDP-bound G12D KRAS is mainly distributed across conformational subspaces. The distances between the Cα atoms of Y32 and Q61 are 23.2, 19.9, 16.2, and 11.3 Å in the structures EB1, EB2, EB3, and EB4 (Appendix A), respectively, which implies that the switch domains of the structure EB1 are the loosest, and the ones of the structure EB4 are the tightest. Compared to the WT KRAS, G12D alters the compact extent of the domain between SW1 and SW2. Based on the superimposition of the structures EB1-EB4 (Figure 6B), it was observed that the SW1 only yields minor deviations, but the loop L1 from the SW2 is extremely out of order. By referencing the WT KRAS, G12D not only induces more energetic states of KRAS, but it also makes the L1 in the SW2 be more disordered. Furthermore, the deviations of the loop L4 and the helix α4 among four structure EB1-EB4 in the G12D KRAS are smaller than that in the WT one (Figure 6B). As displayed in Appendix A, although GDP is aligned well in four structures EB1-EB4, magnesium ions in four structures deviate evidently from each other and are classed into two groups, indicating that magnesium ion is instable through GaMD simulations. The interaction networks identified by the PLIP sever indicate that the salt bridge interaction of GDP with K16, EI of GDP with D119, and the HBIs of GDP with G13, G15, K16, S17, A18, N116, K117, S145, A146, and K147 appear at the most incompact and compact states of the switch domains, respectively, corresponding to the structures EB1 and EB4 (Figure 6C,D). Differently, the HBIs of GDP with V14 and Y32 observed at the structure EB1 are missing at the structure EB4, while the π–π interaction of GDP with F28 and the HBIs of GDP with V29 and D30 appearing at the structure EB4 are lost at the structure EB1 (Figure 6C,D), which reflect the impacts of G12D on the GDP-KRAS binding. Additionally, a sodium ion appears near the phosphate group of GDP in the structure EB4 (Appendix A), implying that the changes in the polarity happen near the phosphate group of GDP due to G12D.

On the GDP-bound G12R KRAS, GaMD simulations recognize four free energy basins, EB1-EB4 (Figure 7A), showing that the GDP-bound G12R KRAS goes across four main conformational subspaces. The distances between the Cα atoms of Y32 and Q61 are 22.6, 22.6, 20.6, and 16.1 Å in the structures EB1, EB2, EB3, and EB4 (Appendix A), individually. Although the switch domains of the structures EB1 and EB2 are located at the most incompact state, the structure EB1 has a bigger structural fluctuation than the structure EB2 (Figure 6A and Appendix A). Differently, the switch domains of the structure EB4 are situated at the tightest state (Appendix A). By comparison with the GDP-bound WT KRAS, G12R leads to a looser switch domain of KRAS and changes the structural compact extent. On the basis of structural alignments of four structures, EB1-EB4 (Appendix A), the switch domains SW1 and SW2 greatly deviate from each other and are located at a extremely disordered state. Compared to the WT KRAS, G12R not only causes more free energy states of KRAS, but it also induces a more disordered state of the switch domains (Appendix A). Meanwhile, the loop L4 and the helix α4 have smaller deviations in the G12R KRAS than in the WT one (Appendix A). Different from the highly disordered states of the switch domains, GDP and magnesium ions are aligned well (Figure 6B), indicating that these two ligands are stable during GaMD simulations. To access the effect of G12R on the GDP-KRAS binding, the interaction network between GDP and the G12R KRAS was analyzed based on the structures EB1 and EB4 by using the PLIP sever (Figure 6C,D). The conserved salt bridge interaction, EI, and HBIs of GDP with residues in the G12R KRAS are similar to those in the GDP-bound WT KRAS. The only difference is that the HBI of GDP with V29 appearing at the structure EB1 disappears at the structure EB4 (Figure 6C,D). More interestingly, a sodium ion appears at all four structures, EB1-EB4, but the position of the sodium ion in the structure EB4 is different from that in the structures EB1-EB3, which embodies the change in the polarity induced by G12R.

According to the aforementioned analyses, three interesting conclusion can be obtained: (1) the sodium ions appear at some conformations of the GDP-bound WT and muted KRAS, implying the change in the polarity near the appearing position of sodium ions; (2) G12A, G12D, and G12R induce more conformational states of the mutated KRAS than those of the WT one and lead to more disordered states of the switch domains; and (3) three G12 mutations alter conformational states of the loop L4 and the helix α4, which is mainly involved in an allosteric position of KRAS and affects the activity of KRAS [26]. The analyses of current EFLs reveal that three mutations lead to conformational transitions between the compact and incompact states of the switch domains in KRAS. Moreover, the changes in the conformational states due to G12 mutations can be used to tune the activity of KRAS. The switch domains mostly overlap with binding regions of KRAS to effectors and nucleotides. Thus, the highly disordered states of the switch domains certainly impact the KRAS–effector binding, which has been revealed by previous studies [25,28,41].

### 2.3. Dynamics Behavior Revealed by PCA

To better study concerted motions of the structure domains, PCA was performed by diagonalizing a covariance matrix built by using the coordinates of the Cα atoms stemming from the single GaMD trajectory (SGT) formed by integrating separate GaMD trajectories. The function of eigenvalues as the eigenvector indexes was depicted in Appendix A. It is noted that the first eigenvalue representing principal concerted motion fast abates to reach more local and minimized conformational space. The first six eigenvectors account for 70.1, 86.4, 73.2, and 88.6% of the total movements of the GDP-bound WT, G12A, G12D, and G12R KRAS, respectively. The first eigenvalue of the three GDP-bound mutated KRAS is greater than that of the GDP-bound WT KRAS, suggesting that G12A, G12D, and G12C strengthen the fluctuation amplitude of KRAS along the first eigenvector relative to the WT KRAS and change dynamics behavior of KRAS.

To catch concerted motions of the structure domains from KRAS, the first eigenvector arising from PCA was visualized by using the VMD software, and the results were depicted in Figure 7. The results not only display highly concerted motions of several domains, but they also verify obvious effect of three G12 mutations on concerted motions of KRAS. For the GDP-bound WT KRAS, the switch domains SW1 and SW2 show strong structural fluctuation. Moreover, the fluctuation tendency of these two structural domains is completely opposite, which makes the SW1 and SW2 go closely to each other (Figure 7A). Besides, the loop L4 yields strong structural fluctuation in the WT KRAS (Figure 7A). By referencing the WT KRAS, G12A, and G12D, this evidently strengthens the structural fluctuation of the SW2. Meanwhile, this also entirely alters the fluctuation tendency of SW1 and SW2, which leads to a tendency of going away from each other (Figure 7B,C). Two mutations, G12A and G12D, also weaken the structural fluctuation of the loop L4 compared to the WT KRAS (Figure 7B,C). By comparison with the WT KRAS, G12R changes the fluctuation tendency of the switch domains and slightly enhances the structural fluctuation of SW1 and SW2 (Figure 7D). Similar to G12A and G12D, G12R also inhibits the fluctuation strength of the loop L4 relative to the WT KRAS.

In summary, G12A, G12D, and G12R produce evident influences on concerted motions of the switch domains from KRAS and change their fluctuation strength and tendency. It is well known that the switch domains of KRAS take part in binding of KRAS to its effectors and regulators [17,36,69]. Thus, the effect of G12 mutations on concerted motions can alter the activity of KRAS. In addition, three G12 mutations also disturb concerted motions of the loop L4 and the helix α4, involved in an allosteric position of KRAS, hence G12 mutations also impact the allosteric regulation on the activity of KRAS [36]

### 2.4. Interaction Network of GDP with KRAS

The previous analyses of FELs revealed that conformational changes induced by G12A, G12D, and G12R alter the GDP–KRAS interaction. To examine the stability of the GDP–KRAS interaction network, the CPPTRAJ program in Amber 20 was used to analyze the information on salt bridge interactions, HBIs, and π–π interactions. The time evolution and probability distributions of distances involved in salt bridge interaction, EI, and π–π interaction were exhibited in Figure 8, and their corresponding structural information was depicted in Figure 9. The HBIs of GDP with KRAS and the geometric positions were displayed in Table 1 and Figure 9. The distances relating to interactions of magnesium ion with residues and GDP and their probability distributions are shown, respectively, in Figure 10 and Figure 11.

The salt bridge interaction of GDP with K16 is identified by the PLIP sever and the time evolution of the distances between the phosphorus atom PB of GDP, and the nitrogen atom NZ was calculated based on the SGT (Figure 8A and Figure 9A). These distances fluctuate from 2.78 to 5.08 Å in four simulated systems (Figure 8A), indicating that this salt bridge interaction is stable through GaMD simulations of four systems. According to the probability distribution of these distances (Figure 8B), G12A and G12R strengthen the salt bridge interaction between GDP and K16, while G12D hardly produces influence on this salt bridge. As displayed in Figure 9A, the phenyl group of F28 is located near the guanine group of GDP. Thus, they easily form the π–π interaction, and Figure 8C provides the time course of the distances between the mass center of the phenyl group of F28 and that of the guanine group of GDP. The distances involved in this π–π interaction are located at a range of 3.89–15.40 Å, signifying that this π–π interaction is basically stable though GaMD simulations. The probability distribution of these distances for this π–π interaction almost overlaps (Figure 8D), implying that three G12 mutations hardly exert effect on this π–π interaction. Figure 9A shows that the guanine group of GDP also forms long-range EI with the carbonyl group of D119, and Figure 8E exhibits the time evolution of the distance between the mass center of three nitrogen atoms, N1-N3, of GDP and that of two oxygen atoms, OD1 and OD2. This distance fluctuates at a range from 5.35 to 7.56 Å in four systems, verifying that the EI is stable during GaMD simulations. The probability distribution of the distance for the salt bridge interaction of GDP with D119 almost overlaps (Figure 8F), suggesting that three G12 mutations hardly affect this long-range EI.

According to Table 1 and Figure 9B, GDP yields the HBIs with residues in the P-loop, including G13, V14, G15, K16, S17, and A18, and the occupancy of the other hydrogen bonds is greater than 83.4%, apart from V14, showing that these hydrogen bonds are stable during GaMD simulations. By referencing the WT KRAS, three G12 mutations decrease the occupancy of the hydrogen bond between GDP and V14. Meanwhile, G12D also reduces that of the hydrogen bonds between GDP and two residues, S17 and A18. Additionally, GDP also generates the HBIs with residues V29 and D40 with an occupancy lower than 34.9%. This result indicates that the HBIs of GDP with SW1 are extremely stable through GaMD simulations (Table 1 and Figure 9B). Compared to the WT KRAS, G12A and G12R obviously decrease the occupancy of the hydrogen bonds between GDP and D30, while G12D raises them. These results provide rational explanation for the high disorder of the switch domains. As observed in Table 1 and Figure 9B, GDP produces the HBIs with two residues (N116 and D119) in loop L4, and their occupancy is higher than 78.1%, verifying that these hydrogen bonds are well kept during GaMD simulations. Although GDP also forms a hydrogen bond with the residue K117 in the loop L4 (Table 1 and Figure 9C), this hydrogen bond is extremely unstable. By comparison with the WT KRAS, G12A and G12D lead to the obvious reduction in the occupancy of two hydrogen bonds between GDP and D119, but G12R hardly impacts the stability of these two hydrogen bonds. In addition, GDP also generate HBIs with three residues, S145, A146, and K147, from the loop L5, and these hydrogen bonds have an occupancy higher than 60.3% (Table 1 and Figure 9C), implying that these hydrogen bonds are well maintained through GaMD simulations. Based on comparison, three G12 mutations scarcely influence the stability of these hydrogen bonds.

To check the stability of magnesium ion during GaMD simulations, the distances of magnesium away from the oxygen atom OG1 of S17, the oxygen atom O of D33, the mass center of the oxygen atoms OD1 and OD2 from D57 and the oxygen atom O3B of GDP were calculated, and their time evolutions and probability distributions were depicted in Figure 9D, Figure 10, and Figure 11. As found in Figure 10A, the distances between magnesium ion and the oxygen atom OG of S17 fluctuate from 1.88 to 2.79 Å in the GDP-bound WT, G12A, and G12R KRAS, but this distance is located a range of 1.91–9.01 Å in the GDP-bound G12D KRAS, which demonstrates that the stability of magnesium ion in the G12D KRAS is much weaker than that in the WT, G12A, and G12R KRAS. The distance of magnesium ion away from the OG of S17 is distributed at a single peak of 2.08 Å in the GDP-bound WT, G12A, and G12R KRAS, while this distance is populated at two peaks of 2.08 and 4.56 Å in the GDP-bound G12D KRAS (Figure 10B). The distances of magnesium ion away from the oxygen atom O of D33 in the SW1 are situated at a range from 3.41 to 19.97 Å (Figure 10C), implying that the position of the SW1 relative to magnesium ion is greatly dynamic. More importantly, these distances are distributed at multiple peaks (Figure 10D). Thus, the position and orientation of the SW1 relative to magnesium ion is highly out of order. The distance between magnesium ion and the mass center of two oxygen atoms OD1 and OD2 of D57 in the SW2 produces a fluctuation range of 3.40–5.81 Å in the WT KRAS, while this distance fluctuates from 2.22 to 3.38 Å in the G12A and G12R KRAS (Figure 11A), indicating that the stability of the magnesium ion in the G12A and G12R KRAS is higher than that in the WT KRAS. Different from the GDP-bound WT, G12A, and G12R KRAS, this distance fluently transforms in three different states in the GDP-bound G12D KRAS (Figure 11A). As shown in Figure 11B, the distance of the magnesium ion away from the mass center of OD1 and OD2 in D57 is located at the peak of 2.79 Å in the G12A and G12R KRAS, situated at the peak of 3.89 Å in the WT KRAS and distributed at three peaks of 4.16, 4.99, and 7.46 Å in G12D KRAS. Thus, the interaction of the magnesium ion with D57 in the WT KRAS is weaker than that in the G12A and G12R KRAS, but it is stronger than that in the G12D KRAS. The distance between the magnesium ion and the oxygen atom O3B of GDP fluctuates at a range of 1.74–2.11 Å in the WT, G12A, and G12R KRAS, while this distance fluently transits between two states in the G12D KRAS (Figure 11C), suggesting that the stability of magnesium ion in the G12D KRAS is much weaker than that in the three other systems. This distance is populated at a single peak of 1.90 Å in the WT, G12A, and G12R KRAS, but it is distributed at two peaks of 1.91 and 3.46 Å in the G12D KRAS, indicating that G12D heavily weakens the interaction of the magnesium ion with GDP.

Based on the aforementioned analyses, three main findings are obtained: (1) the salt bridge interaction of GDP with K16, EI of GDP with D119, and π–π interaction of GDP with F28 and the HBIs of GDP with conserved residues greatly stabilize the GDP-KRAS binding; (2) the high instability in the HBIs of GDP with SW1 is responsible for the disordered state of the switch domains; and (3) G12D produces the most obvious effect on the interactions of magnesium ion with residues S17, D33, and D57, as well as GDP, and the instability in the relative position of magnesium ion to the SW1 also plays an important role in the highly disordered state of the switch domains.

## 3. Theory and Methods

### 3.1. Preparation of Simulated Systems

The crystal structure (5W22) was extracted from protein data bank (PDB) to assign initial atomic coordinates to the GDP-bound WT KRAS [17]. To keep coordinate consistence of the simulated systems, G12 in the 5W22 was, respectively, mutated into A12, D12, and R12 to yield the GDP-bound G12A, G12D, and G12R KRAS. A magnesium ion (Mg^2+^) and the crystal water molecules were kept in the starting models for the GDP-bound WT and mutated KRAS systems. The protonated states of residues in KRAS were examined with the H++ sever, and the rational protonation was assigned to each residue. The missing hydrogen atoms in 5W22 were connected to their corresponding heavy atoms through the Leap module in Amber 20 [90,91]. The ff19SB force field was used to generate the force field parameters of KRAS [92]. The force field parameters of GDP were extracted from the work of Meagher and coworkers [93]. Each KRAS-related complex was immersed in an octahedral periodic box of water with a buffer of 12.0 Å, and the force field parameters of water molecules were produced by means of the TIP3P model [94]. The appropriate numbers of sodium ions (Na^+^) and chlorine ions (Cl^−^) were added at each system in 0.15 M NaCl of salt strength to generate a neutral simulation system, in which the parameters of Na^+^, Cl^−^, and Mg^2+^ were taken from the study of Joung and Cheatham [95,96].

### 3.2. GaMD Simulations

To relieve high-energetic contacts between atoms arising from the initialization of the systems, a five-step solvent minimization of the system was performed with different harmonic force constraints on the solute atoms, such as 100, 50, 10, 5, and 0.0 kcal·mol^−1^·Å^−2^, respectively. Each minimization step consists of 4000 steps of steepest descent minimization and 6000 steps of conjugate gradient minimization. Then, each system was softly enhanced up from 0 to 300 K within 2 ns in a canonical ensemble (NVT) with a weak harmonic restriction of 2 kcal·mol^−1^·Å^2^ on non-hydrogen atoms of the KRAS and GDP. Subsequently, a 2-ns equilibrium process was run at 300 K under the isothermal−isobaric ensemble (NPT). After that, a 20-ns NPT simulation was implemented to keep the density of each simulated system at 1.01 g·cm^−3^. At last, three independent cMD simulations, each for running 300 ns, were performed to relax each system. The initial atom velocities were assigned to three ending structures arising from cMD simulations by using the Maxwell distribution to start three independent GaMD simulations on each system, each running at 1.2 μs.

As for GaMD simulations, a harmonic boost potential is employed to reduce free energy barriers in each system and to obtain more rational conformation samplings. Usually, if $V(\overset{⃑}{r})$ is lower than a threshold energy, E, the potential energy $V(\overset{⃑}{r})$ of a system is changed into $V^{*}(\overset{⃑}{r})$ through Equations (1) and (2), as below:(1)$$V^{*}\left(\overset{⃑}{r}\right)=V\left(\overset{⃑}{r}\right)+∆V(\overset{⃑}{r})$$
(2)$$∆V\left(\overset{⃑}{r}\right)=\left\{\begin{array}{l} 0,\;\;\;\;\;\;\;V(\overset{⃑}{r})\geq E\\ \frac{1}{2}k{\left(E-V\left(\overset{⃑}{r}\right)\right)}^{2},\;V\left(\overset{⃑}{r}\right)<E \end{array}\right.$$
in which the parameter k is the harmonic force constant. Furthermore, the two parameters

E and k can be adjusted by following the criterion in Equations (3) and (4).
(3)$$V_{max}\leq E\leq V_{min}+\frac{1}{k}$$
(4)$$k=k_{0}\frac{1}{V_{max-}V_{min}}$$
If E is set as the lower bound, $E=V_{max}$, then $k_{0}$ is calculated through the following equation:(5)$$k_{0}=\min(1.0,\;\frac{\sigma_{0}}{\sigma_{V}}\cdot \frac{V_{max}-V_{min}}{V_{max}-V_{avg}})$$

To the contrary, if E is given as the upper bound $E=V_{min}+\frac{1}{k}$, then $k_{0}$ is obtained from Equation (6).
(6)$$k_{0}=(1.0-\frac{\sigma_{0}}{\sigma_{V}})\cdot (\frac{V_{max}-V_{min}}{V_{avg}-V_{min}})$$
where the $V_{max}$, $V_{min}$, and $V_{avg}$ involved in the aforementioned equations signify the maximum, minimum, and averaged potential energies of each system, separately, and they were taken from the previous cMD simulations. The parameter $\sigma_{V}$ is the standard deviation of the system potential energies, while the parameter $\sigma_{0}$ is a user-determined upper limit for accurately reweighting. Currently, three independent GaMD simulations, each running for 1.2 μs, were executed on the GDP-bound WT and mutated systems with the periodic boundary conditions. Three separate GaMD trajectories were connected to a single GaMD trajectory (SGT) for facilitating the post-processing analysis. The program PyReweighting provided by Miao and coworkers was adopted to accurately reweight free energies and recognize the native free energy profile of four KRAS-correlated systems [97]. In all cMD and GaMD simulations in this study, the hydrogen-heavy atom chemical bonds were constrained through the SHAKE algorithm [98]. The temperature of each KRAS-correlated system was controlled by means of the Langevin thermostat with a collision frequency of 2.0 ps^−1^ [99]. The particle mesh Ewald (PME) method [100] and an appropriate cutoff value of 12 Å were employed to estimate EIs. Moreover, this cutoff was also adopted to compute the van der Waals interactions. All simulations involved in this study were run by using the program pmemd.cuda inlayed in Amber 20 [101,102].

### 3.3. Principal Component Analysis

To explore concerted motions coupling with functional significance, PCA was executed in this study. The first step of PCA is to build a covariance matrix C of the C_α_ atoms on the basis of the following equation:(7)$$C=<(q_{i}-<q_{i}>){\left(q_{j}-<q_{j}>\right)}^{T}>$$

In which $q_{i}$ and $q_{j}$, respectively, are the Cartesian coordinates of the ith and *j*th C_α_ atoms in KRAS, while $<q_{i}>$ and $<q_{j}>$ reflect their averaged positions over conformational ensembles saved at the SGT. In general, the average is estimated by aligning the SGT with a referenced structure to abolish overall translations and rotations by using a least-square fit procedure [103]. The second step of PCA is to diagonalize the symmetric matrix C to yield a diagonal one A with an orthogonal coordinate transformation matrix T through the following equation:(8)$$A=T^{T}C_{ij}T$$
in which the diagonal elements of A are the eigenvalues $\lambda_{i}$ and the columns of A correspond to the eigenvectors, reflecting the motion direction relative to $<q_{i}>$. The third step of PCA is to explore concerted movements of structural domains in a multidimensional space using the eigenvector and to describe the fluctuation amplitude along an eigenvector, which can efficiently characterize collective motions of structural domains from KRAS. In this study, PCA was completed by employing the CPPTRAJ module in Amber 20 [104]. The software VMD was wielded to finish visualization of the eigenvectors stemming from the PCA [105], to depict pictures, and to reveal influences of G12 mutations on concerted motions of the structural domains in KRAS. 

### 3.4. Dynamics Cross-Correlation Maps

To understand effect of G12 mutations on internal dynamics of structural domains in KRAS, the elements $C_{ij}$ of DCCMs were calculated by means of the x, y, and z coordinates of the C_α_ atoms in the backbone of KRAS on the basis of Equation (9):(9)$$C_{ij}=\frac{<{∆r}_{i}\cdot {∆r}_{j}>}{{\left(<{∆r}_{i}^{2}><{∆r}_{j}^{2}>\right)}^{1/2}}$$
in which ${∆r}_{i}$ and ${∆r}_{j}$ separately represent the displacement of atoms i and j relative to their corresponding averaged positions. The angle brackets signify ensemble averages over the snapshots recorded at the SGT. The element values of DCCMs fluctuate from −1 to 1. The positive and negative $C_{ij}$ values, respectively, describe the positively correlated motions and the anti-correlated movements between the C_α_ atoms *i* and *j*. The color-coded styles were applied to represent the extent of correlated motions. The module CPPTRAJ in Amber 20 was also utilized to realize computations of DCCMs.

## 4. Conclusions

G12 mutations produce significant effects on the conformational states of KRAS and are involved in the development of human cancers. Insights into mutation-induced conformational changes of KRAS are of great significance for further clarifying molecular mechanism of the KRAS activity regulation. The 3.6-μs GaMD simulation, composed of three independent GaMD simulations of 1.2 μs, was performed on the GDP-bound WT, G12A, G12D, and G12R KRAS to probe the effect of mutations on conformational transformations of KRAS. The calculated RMSFs and DCCMs verify that G12A, G12D, and G12R alter the structural flexibility and internal dynamics of the switch domains. The FELs constructed by using the RMSDs of non-hydrogen atoms from KRAS and the distance of Y32 away from Q61 as coordination coordinates show that three G12 mutations not only lead to more conformational states of KRAS, but they also make the switch domains more disordered. The PCA verifies that G12A, G12D, and G12R exert significant effect on collective domains of the switch domains and change the fluctuation amplitude. The great changes in the order extent of the switch domains that mostly overlap with the binding regions of KRAS to its effectors certainly affect the activity of KRAS. The identified interaction networks further reveal that the high instability in the HBIs of GDP with residues V29 and D30 in the SW1 are mostly responsible for the extremely disordered state of the switch domains. This work can contribute useful aids to deep insights into the function of KRAS.

## Figures and Tables

**Figure 1 molecules-28-02886-f001:**
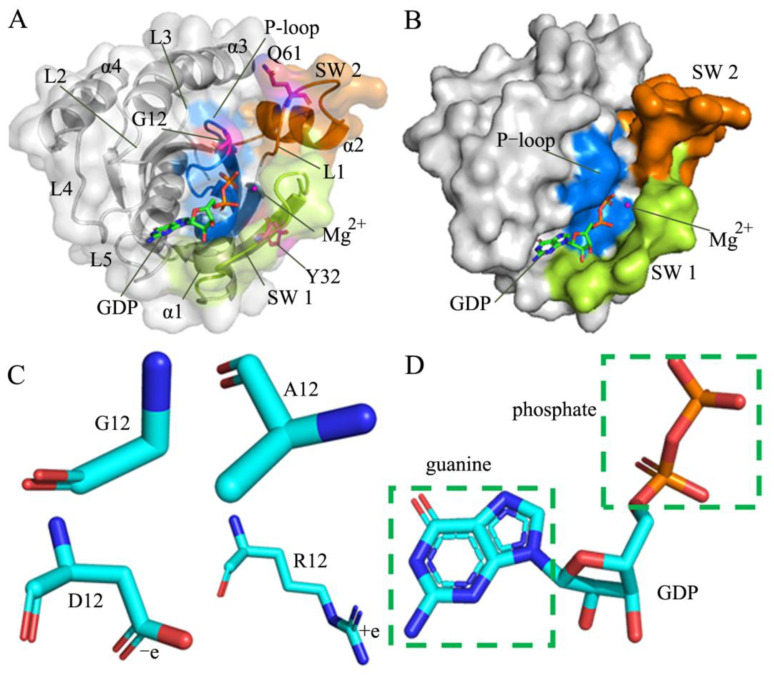
Structural information: (**A**) the GDP/WT KRAS complex, in which KRAS is shown in surface and cartoon modes while GDP, magnesium ion, and key residues are displayed in stick, ball, and stick modes, respectively, (**B**) binding sites of GDP to KRAS displayed in surface styles, (**C**) changes of structure and polarity or hydrophobicity due to mutations of G12 into A12, D12, and R12, in which key residues shown in stick modes and (**D**) GDP displayed in stick patterns. In Figure 1A and 1B, the blue, limon and orange respectively indicate the P-loop, SW 1 and SW 2.

**Figure 2 molecules-28-02886-f002:**
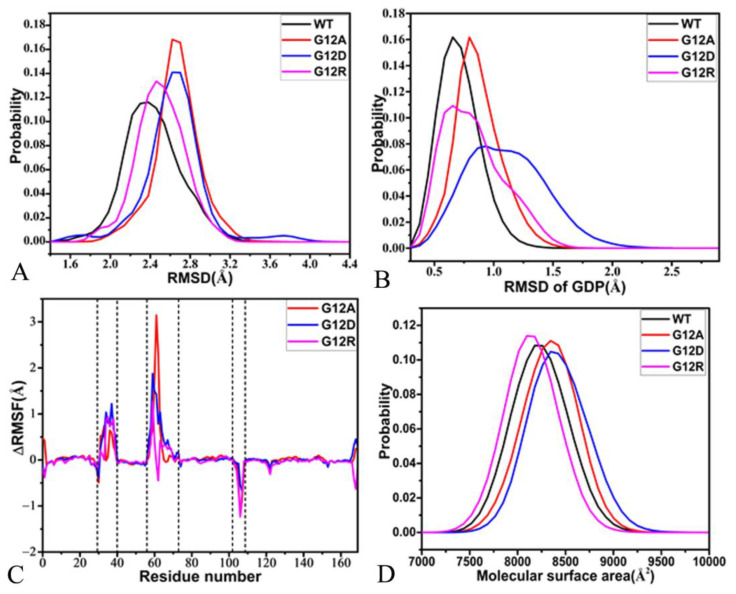
Structural fluctuations and flexibility: (**A**) probability of RMSDs for non-hydrogen atoms of KRAS, (**B**) probability of RMSDs for non-hydrogen atoms of GDP, (**C**) difference in RMSFs of the Cα atoms from KRAS, and (**D**) molecular surface areas of KRAS.

**Figure 3 molecules-28-02886-f003:**
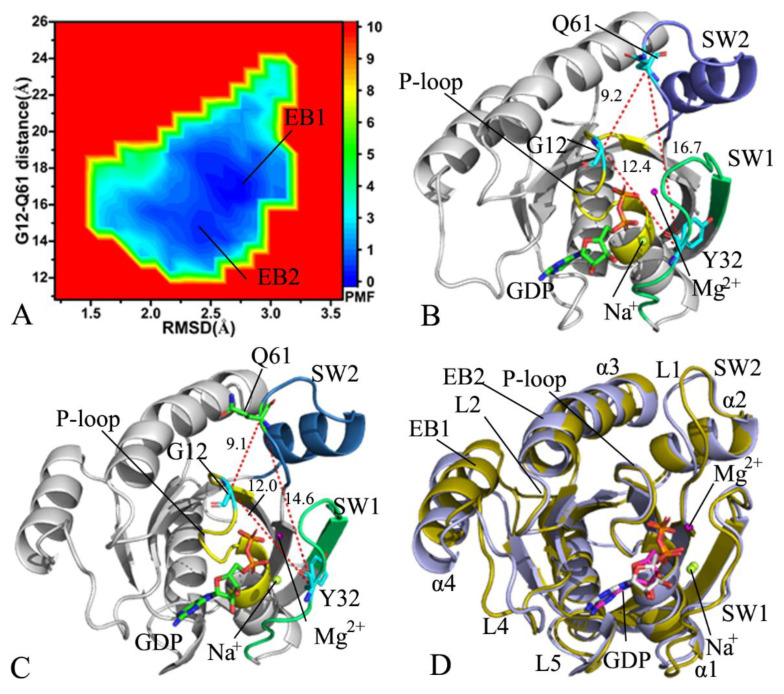
Free energy profiles and structural information of the GDP-bound WT KRAS: (**A**) the FEL built by using the RMSD and the distance of Y32 away from Q61, (**B**) the structure in the EB1, (**C**) the structure in the EB2, and (**D**) superimposition of the structures for KRAS in the EB1 and EB2. The distances are scaled in Å, and the PMF is indicated in kcal/mol.

**Figure 4 molecules-28-02886-f004:**
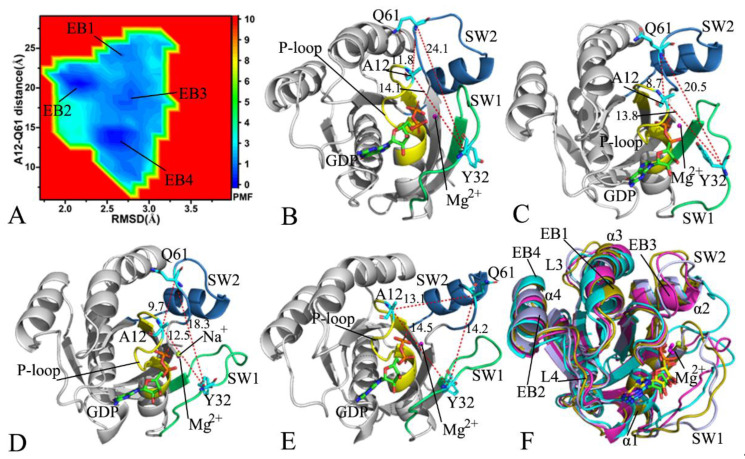
Free energy profile of structural information of the GDP-bound G12A KRAS: (**A**) the FEL built by using the RMSD and the distance of Y32 away from Q61, (**B**) the structure in the EB1, (**C**) the structure in the EB2, (**D**) the structure in the EB3, (**E**) the structure in the EB4, and (**F**) the structural alignment of the G12A KRAS in the EB1-EB4. The distances are scaled in Å, and the PMF is indicated in kcal/mol.

**Figure 5 molecules-28-02886-f005:**
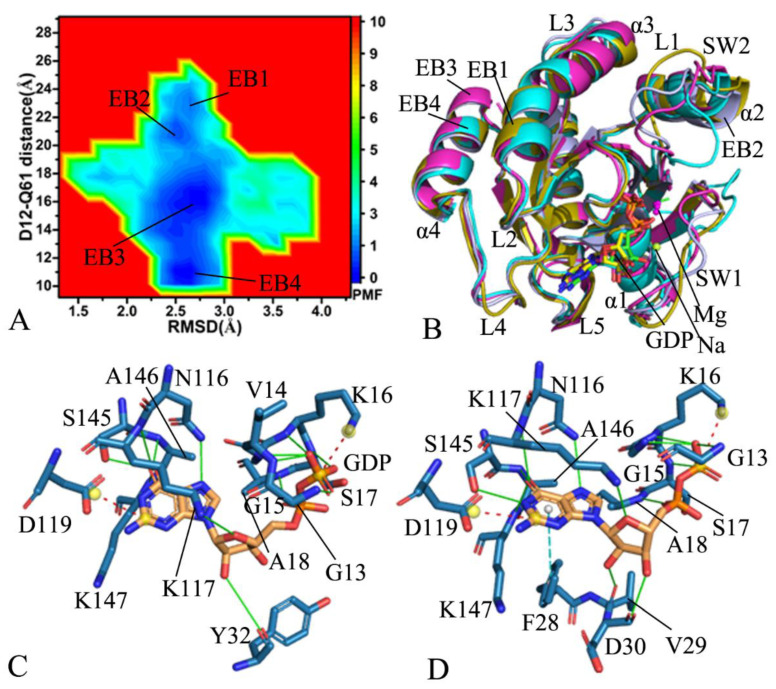
Free energy profile of structural information of the GDP-bound G12D KRAS: (**A**) the FEL built by using the RMSD and the distance of Y32 away from Q61, (**B**) superimposition of structures for KRAS in the EB1-EB4, (**C**) the GDP-residue interaction in the most incompact state of the switch domains, and (**D**) the GDP–residue interaction in the tightest state of the switch domains. The PMF is scaled in kcal/mol.

**Figure 6 molecules-28-02886-f006:**
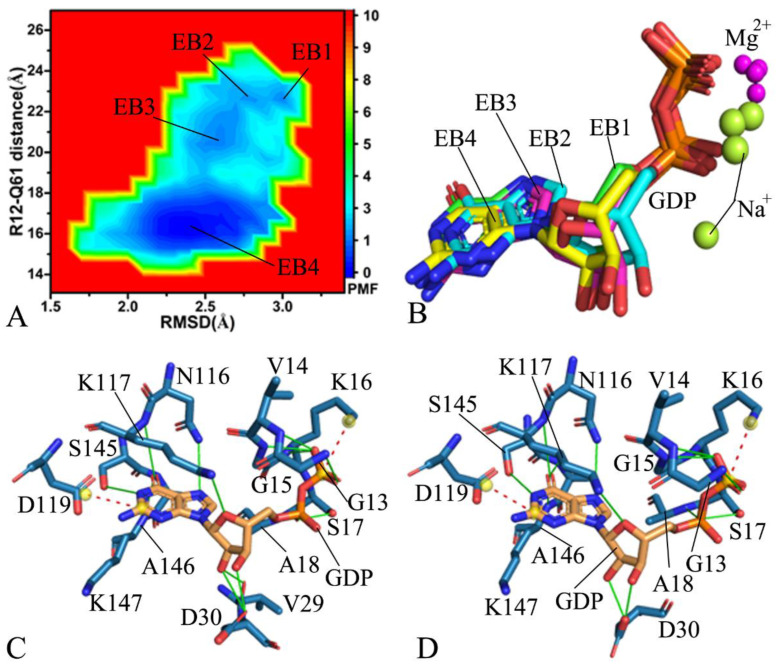
Free energy profile of structural information of the GDP-bound G12R KRAS: (**A**) the FEL built by using the RMSD and the distance of Y32 away from Q61, (**B**) superimposition of structures for GDP and magnesium ions in the EB1-EB4, (**C**) the GDP–residue interaction in the most incompact state of the switch domains, and (**D**) the GDP–residue interaction in the tightest state of the switch domains. The PMF is scaled in kcal/mol.

**Figure 7 molecules-28-02886-f007:**
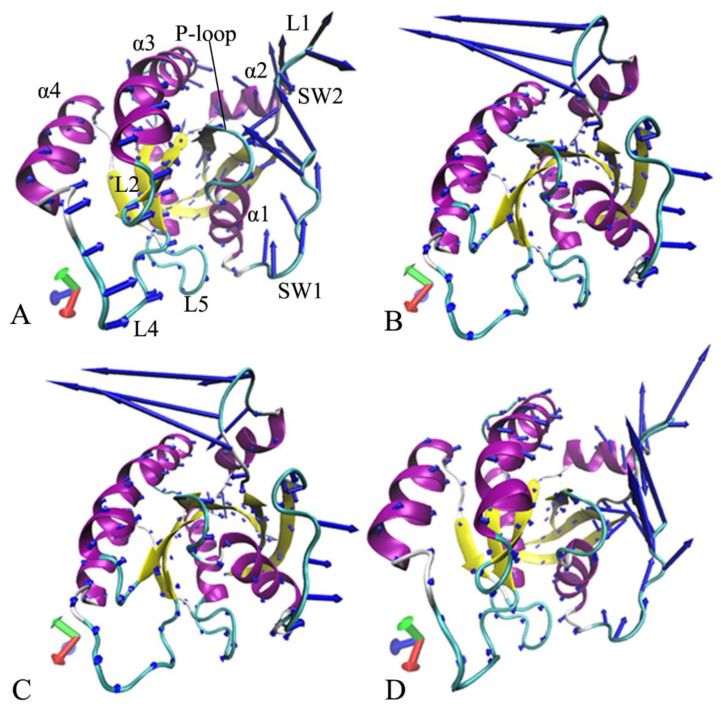
Concerted motions of the structural domains revealed by the first eigenvector arising from PCA: (**A**) the GDP-bound WT KRAS, (**B**) the GDP-bound G12A KRAS, (**C**) the GDP-bound G12D KRAS, and (**D**) the GDP-bound G12R KRAS. The violet, blue and cyan separately represent the helix, β-sheet and loop.

**Figure 8 molecules-28-02886-f008:**
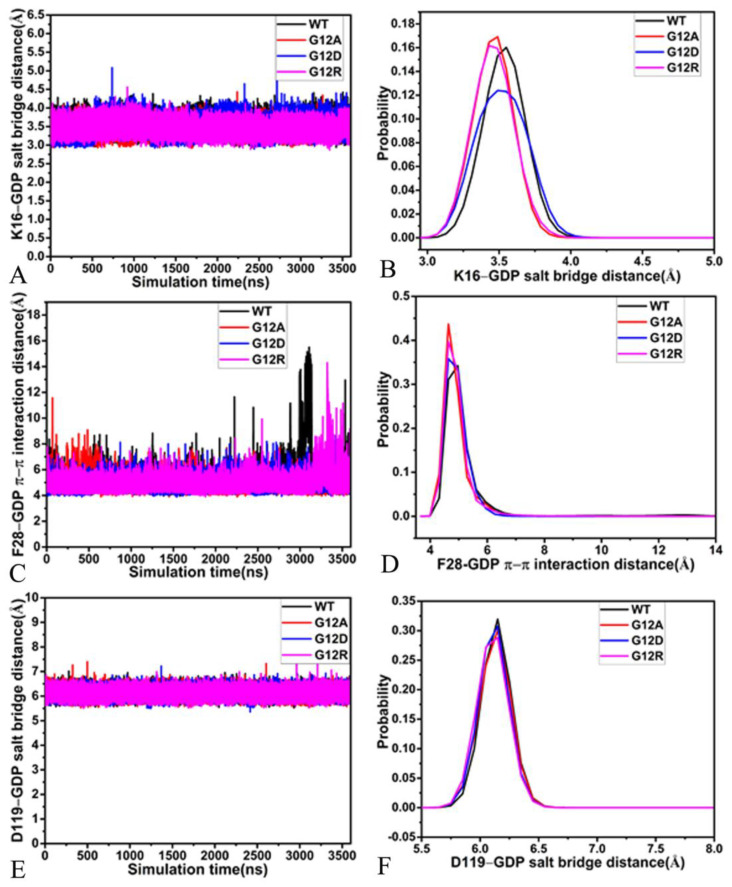
The time evolution of distances and their probability distributions: (**A**,**B**) separately corresponding to the salt bridge interaction of GDP with K16 and its probability distribution, (**C**,**D**) respectively, describing the distance of the π–π interaction of GDP with F28 and its probability distributions, (**E**,**F**) respectively, corresponding to the distance of the salt bridge interaction of GDP with D119 and its probability distributions.

**Figure 9 molecules-28-02886-f009:**
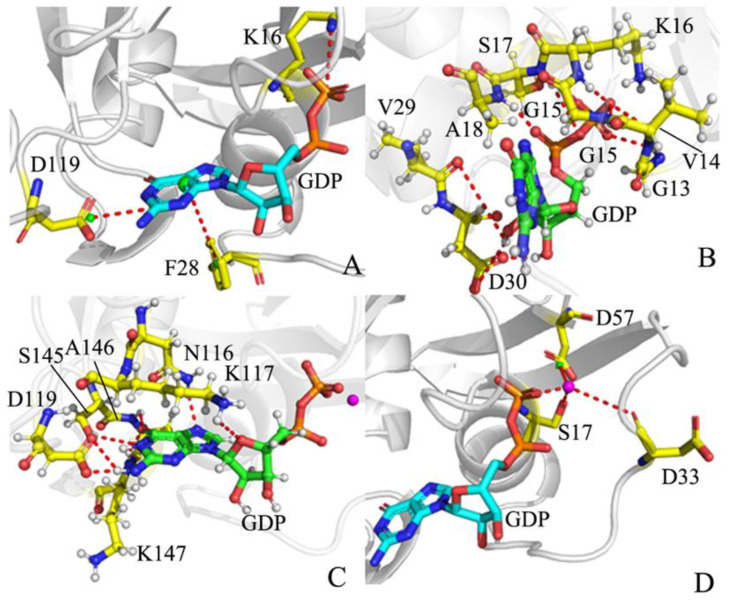
Geometric position of GDP relative to key residues and magnesium ion: (**A**) the salt bridge interactions of GDP with K16 and D119, as well as the π–π interaction of GDP with F28, (**B**) the HBIs of GDP with residues in the P-loop and SW1, (**C**) the HBIs of GDP with residues in the loop L4 and L5, and (**D**) interactions of the magnesium ion with residues and GDP.

**Figure 10 molecules-28-02886-f010:**
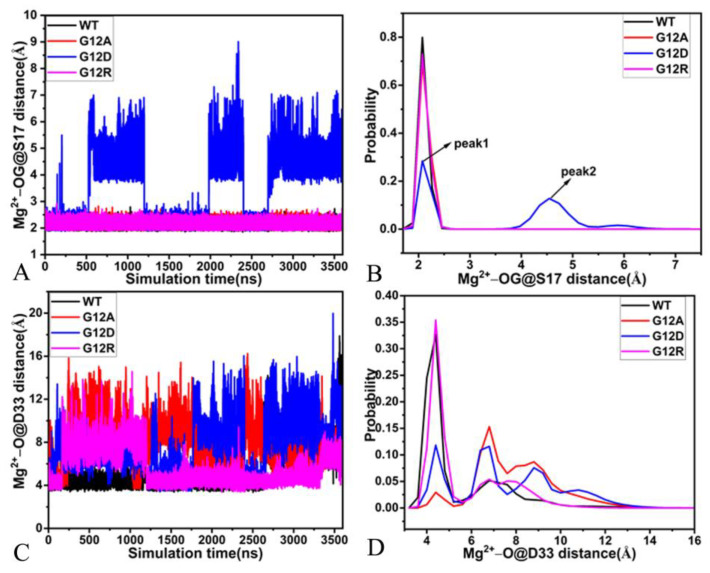
The time evolution and probability distributions of distances between magnesium ion and residues S17 and D33: (**A**,**B**) corresponding to the time evolution of the distance between magnesium ion and the oxygen atom OG of S17 and its probability distribution, respectively, and (**C**,**D**) corresponding to the time course of the distance between the magnesium ion and the oxygen atom O of D33 and its probability distributions, individually.

**Figure 11 molecules-28-02886-f011:**
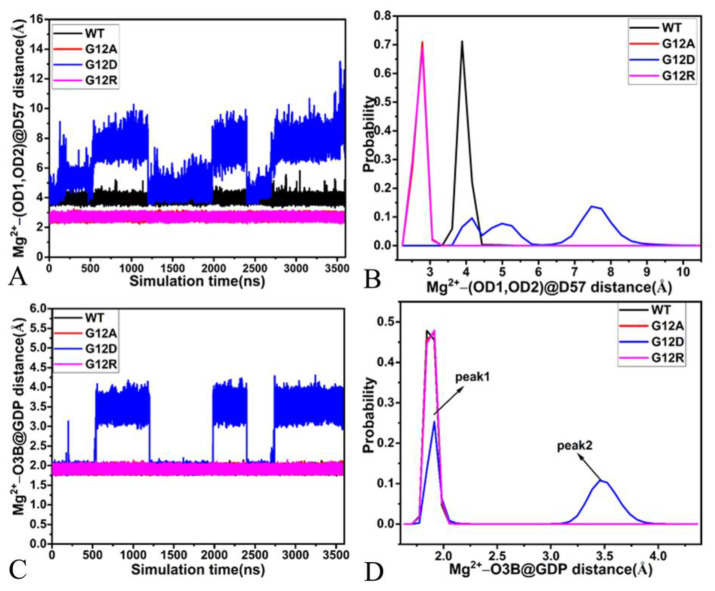
The time evolution and probability distributions of distances between magnesium ion and residues D57 and GDP: (**A**,**B**) corresponding to the time evolution of the distance between magnesium ion and the mass center of the oxygen atoms OD1 and OD2 of D57 and its probability distribution, respectively, and (**C**,**D**) corresponding to the time course of the distance between magnesium ion and the oxygen atom O3B of GDP and its probability distributions, individually.

**Table 1 molecules-28-02886-t001:** HBIs of GDP with KRAS revealed through the program CPPTRAJ.

Hydrogen Bonds ^a^		Occupancy (%) ^b^			
Residue	GDP	WT	G12A	G12D	G12R
K16-N-H	O2B	99.9	99.9	99.2	99.9
A18-N-H	O2A	99.4	98.5	83.4	99.2
G15-N-H	O2B	98.8	99.2	99.2	99.1
S17-N-H	O3A	98.1	99.8	85.6	99.6
G13-N-H	O1B	96.8	97.7	92.7	98.7
V14-N-H	O2B	41.2	30.6	32.3	30.1
N116-ND2-HD21	N7	89.4	93.1	85.9	92.8
K117-NZ-HZ2	O4’	20.3	16.9	11.8	23.7
S145-OG-HG	N1	64.1	60.3	61.5	60.5
K147-N-H	O6	86.8	85.2	83.9	87.1
A146-N-H	O6	60.8	63.1	67.3	61.7
D119-OD1	N1-H1N	97.6	80.1	88.5	95.1
D119-OD2	N2-H21	95.6	83.6	78.1	92.4
V29-O	O2’-H2’	25.5	22.9	28.8	27.3
D30-OD1	O2’-H2’	19.6	10.3	25.4	10.9
D30-OD2	O3’-H3’	17.8	8.2	23.1	9.2

^a^ HBIs are recognized by acceptor–donor distance of <3.5 Å and acceptor–H-donor angle of >120°. ^b^ Occupancy (%) is defined as the percentage of simulation time that a specific hydrogen bond exists.

## Data Availability

Not applicable.

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
