# Peer review of "Impacts of Mutations in the P-Loop on Conformational Alterations of KRAS Investigated with Gaussian Accelerated Molecular Dynamics Simulations"

_molecules, 2023, doi:10.3390/molecules28072886_

Round 1

Reviewer 1 Report

Please see attached report

Author Response

Reviewer #1

  1. 1.Please recall the acronyms of RAS, KRAS, HRAs and NRAS.

Lines 27-29 should be rewritten.

Reply:

Thank you very much for this valuable suggestion.

Based on your suggestion, we rewritten Lines 27-29 and the revised parts are highlighted in the red.

Thank you again for your comment.

  1. Lines 121-124. Rmsd values of 0.14 and 0.34 are very small. Why then state that the G12A and G12D mutations weaken the stability of GDP in the binding pocket?

Reply:

Thank you very much for this valuable suggestion.

The RMSDs of GDP are increased due to mutations G12A and G12D relative to the WT system, which strengthens the structural fluctuation of GDP in binding pocket of the G12A and G12D KRAS. Thus, the structural stability of GDP in the G12A and G12D KRAS is correspondingly weakened. By following your valuable suggestion, we have revised our manuscript and the revised contents are highlighted in the red.

Thank you again for this valuable suggestion.

  1. Line 4. What is meant by ‘the probability of RMSD for GDP?’

Lines 139-145 are unclear. The DRMSF are not well explained in relation to Figure 2C. Figure 2C is unclear. How can a simulation time on the x axis be related to SW1 in the 30-40ns time interval, then to SW2 in the 55-70 ns interval, then to L3 in the 100-110 ns time interval?

Reply:

Thank you very much for this valuable suggestions, meanwhile we are sorry for our mistake in Figure 2C.

Firstly, the x-axis in Figure 2C should be “Residue number”, we have revised Figure 2C.

Secondly. On the probability, the range from the minimum to maximum values of RMSDs is separated into many intervals, and the probability indicate a percentage of the number located at an interval accounting for the total conformational numbers.

Based on your valuable suggestion, we have revised our manuscript and the revised parts are marked with the red.

Thank you again for these valuable suggestions.

  1. P. 5. Lines 183-186. Could the authors briefly comment the implications of the two sentences for drug design? Is the effector GDP or GTP, or possibly either one?

Reply:

Thank you very much for this valuable suggestion.

By following your suggestion, we have revised our manuscript and the revised parts are highlighted in the red.

  1. P. 6.Line 215, but also throughout the whole paper. The interaction of D119 with GDP is by no means a ‘salt bridge’. It is an H-bond between one anionic O of the formate end side-chain and a NH group of guanine, either –H2N2 or –HN1

Line 235. The word ‘vital’ is inadequate.

Reply:

Thank you very much for these valuable suggestions. We agree with you.

The negative charge in the carbonyl of D119 and the positive charge in the guanine group of GDP can generate the strong long-range electrostatic interaction, which is identified by the PLIP server. Meanwhile, the analysis from the CPPTRAJ suggests that two hydrogen bonding interactions are formed between D119 and the N1H1N and N2-H21 of the guanine group from GDP (Table 1). D119 not only generates hydrogen bonding interactions but also forms strong long-range electrostatic interaction with guanine group of GDP. Based on your valuable suggestions, we have revised our manuscript and the revised parts are marked with the red. Meanwhile, the word “vital” is deleted.

  1. 6. P. 7.Line 252. Could the ‘change of the polarity near the phosphate group’ be given more details?

Line 264. Please replace ‘exhibits’ by another verb.

Reply:

Thank you very much for these valuable suggestion.

Based on your valuable, we have revised our manuscript and the revised contents are highlighted in the red.

Meanwhile, the “exhibits” has been replaced with “implies”.

Thank you again for these valuable suggestions.

  1. P. 11.Line 369. Replace ‘vital’ by another adjective.

Lines 405-406. This sentence is incomprehensible. At first sight, it would appear that 5.4-7.6 A are in a range of long distances for stable interactions. And these are by no means (please see above remark) ‘salt bridge’ interactions.

Reply:

Thank you very much for this valuable suggestion. We also agree with you.

The negative charge in the carbonyl of D119 and the positive charge in the guanine group of GDP can generate the strong long-range electrostatic interaction, which is identified by the PLIP server. Meanwhile, the analysis from the CPPTRAJ suggests that two hydrogen bonding interactions are formed between D119 and the N1H1N and N2-H21 of the guanine group from GDP (Table 1). D119 not only generates hydrogen bonding interactions but also forms strong long-range electrostatic interaction with guanine group of GDP. Based on your valuable suggestion, we have revised Figure 8 and the corresponding information. The revised contents are highlighted in the red.

Meanwhile, The word “vital” is replaced with “evident”.

Thank you again for this valuable suggestion.

  1. 8. 12.Lines 407-409. The words ‘salt bridge’ is again misleading, unless there were a Mg(II) cations chelated by the end side-chain of D119 and a phosphate group of GDP.
  2. 16, lines 500-502. See above remarks about the salt bridge.

Line 519. Replace filed by field.

Reply:

Thank you very much for this valuable suggestion.

By following your valuable suggestion, the information with regard to the salt bridge between D119 and GDP has been revised, which is highlighted in the red.

Meanwhile, the “filed” has been replaced by the “field”

  1. 9. P. 17Line 444. What is the meaning of k’?

Equation 6. How are Vmax, Vmin, and Vag computed? Is this from prior MD simulations prior to restart by GAMD?

Equation 7. How are qi and qj computed? Should not a summation over the explored configurations be set?

Line 583. What is the acronym of ‘SGT’?

Reply:

Thank you very much for these valuable comments and suggestions.

The parameter “k” is the harmonic force constant, which is defined in Miao’s work.

The Vmax, Vmin, and Vag are taken from the cMD simulations prior to restart by GAMD.

The qi and qj are the Cartesian coordinates and they are taken from the GaMD trajectory, while <qi> and <qj> are the average over conformations saved the SGT.

The “SGT” is “a single GaMD trajectory”, which has been provided in the previous part.

Based on your valuable comment, we have revised our manuscript and the revised parts are highlighted in the red.

  1. P. 18Line 614. ‘The development of human oncology’ is an inappropriate wording. Alternatives could be ‘human oncology’ or ‘the development of human cancers’, etc.

Line 623. What is the meaning of ‘the more energetic states of KRAS’?

Line 628. What is meant by ‘the highly instability in the interactions of GDP and Mg(II)? The interactions between GDP and Mg(II) are always stable, albeit at varying extents.

Reply:

Thank you very much for these valuable suggestion.

‘The development of human oncology’ has been revised as ‘the development of human cancers’.

‘the more energetic states of KRAS’? has been revised as ‘more conformation states’.

‘the highly instability in the interactions of GDP and Mg(II)?’ has been revised as “the highly instability in the HBI interactions of GDP and magnesium ion with residues V29 and D30 in the SW1”.

Finally, thank you very much for your valuable suggestions and efforts paid by you in reviewing our manuscript.

Reviewer 2 Report

In this manuscript, the authors study the structures and dynamics of the complexes of KRAS with GDP by means of Gaussian accelerated MD and they conclude that the 12-Gly mutants, mutation in P loop, exhibit the different properties of structures and dynamics compared with the WT complex, especially flexibility in P loop. Thus, the structure and dynamics of P loop will be involved in the function of P loop. The results and conclusions derived by authors are interesting and this paper will be worth publishing after the following points are reconsidered.

I am interested in the results when the authors apply their methods to a mutant of KRAS at near 12-Gly which is confirmed by experiment not to show any reduction of function. If such experimental data is available, the authors should examine it as a control. Then, deeper insight on the function can be obtained.

Minor points;

The key residue numbers, i.e., G12, Y32, Q61 and so on, should be indicated in Figure 1.

In page 3, line 80, “aMD” shoule be (aMD).

In page 3, line 80, “GaMD” appears first here in the main text. So, the complete expression of “GaMD” should be shown here although its full name already appears in the title.

In Figure 3A, the basins EB1 and EB2 are not clear. I cannot identify as two basins. The resolution of the figure should be improved. (The basins in Figure 4A can be identified.)

In page 7, line 238, is the meaning of “energetic state” “high energy state”? Is “energetic state” used commonly?

As the similar word in page 10, line 328, what the meaning of “more free energy state”?

In page 10, line 339, “SGT” appears first here. The complete expression of “SGT” should be shown here.

In page 16, line 526, Na+, Cl- and Mg2+ should be Na^+, Cl^-, and Mg^2+.

Author Response

Reviewer #2:

Thank you very much for your valuable suggestion and comments in advance.

  1. The key residue numbers, i.e., G12, Y32, Q61 and so on, should be indicated in Figure 1.

Reply:

Thank you very much for your valuable suggestion.

Figure 1 has been revised and the corresponding description has been also revised, which is highlighted in the red.    

  1. In page 3, line 80, “aMD” shoule be (aMD).

Reply:

Thank you very much for this kindly reminding.

   The “aMD” has been revised as “(aMD)”.

  1. In page 3, line 80, “GaMD” appears first here in the main text. So, the complete expression of “GaMD” should be shown here although its full name already appears in the title.

Reply:

Thank you very much for this kindly reminding

The full name of GaMD has been added, which is marked in the red.

  1. In Figure 3A, the basins EB1 and EB2 are not clear. I cannot identify as two basins. The resolution of the figure should be improved. (The basins in Figure 4A can be identified.).

Reply:

Thank you very much for this valuable comment. 

According to the color bar, an energy barrier of about 1.5 kcal/mol exists between the basins EB1 and EB2. Based on your suggestion, we redraw figure 1A and improved the resolution.

Thank you again for this valuable suggestion.

  1. In page 7, line 238, is the meaning of “energetic state” “high energy state”? Is “energetic state” used commonly?.

Reply:

Thank you very much for this valuable suggestion.

We have revised the description. The “two energetic states” has been revised as “two structures located at the EB1 and EB4”, which is marked with the red.

  1. 6. As the similar word in page 10, line 328, what the meaning of “more free energy state”?

Reply:

Thank you very much for this valuable comment.

The “more free energy state” has been revised as “more conformational states.”

  1. In page 10, line 339, “SGT” appears first here. The complete expression of “SGT” should be shown here.

Reply:

Thank you very much for kindly reminding.

The full name of SGT has been added, which is marked with the red.

  1. 8.In page 16, line 526, Na+, Cl- and Mg2+ should be Na^+, Cl^-, and Mg^2+.

Reply:

Thank you very much for this kindly reminding.

The mistakes have been corrected.

Finally, thank you very much for your valuable suggestions and efforts paid by you in reviewing our manuscript.

Round 2

Reviewer 2 Report

The authors have revised their manuscript along my suggestion except the following comment;

"I am interested in the results when the authors apply their methods to a mutant of KRAS at near 12-Gly which is confirmed by experiment not to show any reduction of function. If such experimental data is available, the authors should examine it as a control. Then, deeper insight on the function can be obtained."

I hope that the authors give some sentences on this comment although I leave it to the authors whether the authors do it or not. 

Author Response

Dear reviewers:

Thank you for your letter and for the reviewers’ comments concerning our manuscript entitled “Impacts of mutations in the P-loop on conformational alterations of KRAS investigated with Gaussian accelerated molecular dynamics simulations”. The comments are all valuable and very helpful for revising and improving our paper, as well as provide the important guidance for revision of our work. We have investigated the comments carefully and made corrections that we hope meet with approval. Revised portions have been marked in red in our manuscript. The main corrections in the manuscript and the point-to-point responds to the reviewers’ comments are as following:

Reviewer #2

Thank you very much for your valuable suggestions

  1. "I am interested in the results when the authors apply their methods to a mutant of KRAS at near 12-Gly which is confirmed by experiment not to show any reduction of function. If such experimental data is available, the authors should examine it as a control. Then, deeper insight on the function can be obtained.".

Reply:

Thank you very much for this valuable suggestion.

The high flexibility of the switch domains in KRAS is functionally significant and the changes in the flexibility of the switch domains caused by residue mutations certainly produce important effect on the activity of KRAS. Thus, the G12 mutations in the switch domain SW 1 can be applied to regulate the activity of KRAS.

According to your valuable suggestion, we have revised our manuscript and the revised parts are highlighted in the red.

“Therefore, the changes in the structural flexibility and internal dynamics of the switch domains induced by three G12 mutations certainly influence KRAS-effector binding. It is well known that high flexibility is a main feature of the switch regions of RAS proteins, which enables conformational transformation associated with a GDP/GTP exchange [31]. Thus, the changes in the flexibility of the switch domain SW 1 can be applied to regulate the activity of KRAS.”

“The analyses of current EFLs reveal that three mutations lead to conformational transitions between the compact and incompact states of the switch domains in KRAS, moreover the changes in the conformational states due to G12 mutations can be used to tune the activity of KRAS.”

Finally, thank you very much for your valuable suggestions and efforts paid by you in reviewing our manuscript.
